# Mutual Learning for SAM Adaptation: A Dual Collaborative Network Framework for Source-Free Domain Transfer

Yabo Liu [1]  Waikeung Wong [2][3]  Chengliang Liu [2]  Xiaoling Luo [4]  Yong Xu [1]  Jinghua Wang [1]

## Abstract

Segment Anything Model (SAM) has demonstrated remarkable zero-shot segmentation capabilities across various visual tasks. However, its performance degrades significantly when deployed in new target domains with substantial distribution shifts. While existing self-training methods based on fixed teacher-student architectures have shown improvements, they struggle to ensure that the teacher network consistently outperforms the student under severe domain shifts. To address this limitation, we propose a novel Collaborative Mutual Learning Framework for source-free SAM adaptation, leveraging dual-networks in a dynamic and cooperative manner. Unlike fixed teacher-student paradigms, our method dynamically assigns the *teacher* and *student* roles by evaluating the reliability of each collaborative network in each training iteration. Our framework incorporates a dynamic mutual learning mechanism with three key components: a direct alignment loss for knowledge transfer, a reverse distillation loss to encourage diversity, and a triplet relationship loss to refine feature representations. These components enhance the adaptation capabilities of the collaborative networks, enabling them to generalize effectively to target domains while preserving their pre-trained knowledge. Extensive experiments on diverse target domains demonstrate that our proposed framework achieves state-of-the-art adaptation performance. Our code is accessible at https://github.com/yaboliudotug/DMLDA.

## 1. Introduction

The development of deep learning technology has brought remarkable breakthroughs to the field of computer vision (He et al., 2016; Liu et al., 2022; Redmon et al., 2016; Carion et al., 2020; Liu et al., 2025; Sun et al., 2024; Huang et al., 2022b;a). Among these advancements, progress in pre-training techniques has been particularly remarkable, with models like Segment Anything Model (SAM) (Kirillov et al., 2023) demonstrating powerful capabilities across various visual perception tasks(Oquab et al., 2023; Bai et al., 2024). Taking advantage of its pre-training procedure on extensive and diverse datasets, SAM demonstrates remarkable generalization capabilities and delivers high-quality segmentation results. However, SAM often suffers significant performance degradation when deployed in a target domain (*i.e.,* the testing data) that shifts significantly from the source domain (*i.e.,* the training data) (Zhang et al., 2024; Chen et al., 2023; 2024). These domain shifts arise due to differences in visual characteristics, object distributions, or environmental conditions between the pre-training datasets and the target domains. For example, SAM fails to perform well in professional applications, such as medical imaging, camouflaged object detection, and robotic vision. This highlights the urgent need to adapt SAM to target domains for improved usability in real-world applications.

It is infeasible to adapt SAM in the traditional framework of unsupervised domain adapation (UDA), as the large scale pre-training dataset could introduce substantial computational burden and make the adaption procedure both inefficient and unscalable. In addition, the source domain data is not always accessible due to privacy concerns, storage limitations, or data-sharing restrictions. Source-free domain adaptation (SFDA) has emerged as a promising solution to address these issues. SFDA enables the adaptation of a pre-trained model using only unlabeled target domain data, making it particularly applicable to sensitive fields such as medical imaging and autonomous driving. SFDA of SAM requires the model to learn robust representations of the target domain while avoiding catastrophic forgetting of its pre-trained generalization capabilities.

A common approach to SFDA is self-training through teacher-student knowledge distillation. In this paradigm,

[1]Harbin Institute of Technology, Shenzhen [2]Laboratory for Artifcial Intelligence in Design, Hong Kong [3]School of Fashion and Textiles, Hong Kong Polytechnic University [4]Shenzhen University. Correspondence to: Waikeung Wong <calvinwong@aidlab.hk>.

*Proceedings of the 42$^{nd}$ International Conference on Machine Learning*, Vancouver, Canada. PMLR 267, 2025. Copyright 2025 by the author(s).

a teacher network generates pseudo-labels for the target domain data, which are then used to supervise the training of a student network (Zhang et al., 2024). This approach assumes that the teacher consistently provides more reliable predictions than the student. However, this assumption often does not hold in practice, especially when the model is deployed in a new target domain. Our experiments and analysis reveal that the student network may surpass the teacher network in capturing domain-specific features during training. The fixed knowledge transfer direction in the teacher-student frameworks limits the potential of self-training, ultimately leading to suboptimal performance.

To address these limitations, we propose a novel *Collaborative Mutual Learning* framework that replaces the fixed teacher-student paradigm with a more flexible and dynamic one, inspired by (Zhou et al., 2023a). Two networks collaborate in equal roles in the proposed method, and they dynamically learn from each other rather than adhere to a static teacher or student role. This bidirectional knowledge exchange allows both networks to leverage the strengths of the other to achieve superior performance in the target domain.

A key challenge in implementing mutual learning lies on the determination of the relative reliability of the networks during training. To address this, we introduce a dynamic role assignment mechanism based on a knowledge-preserving metric. This metric compares the foreground features of each network with those extracted from the original pre-trained SAM model, and tells how much the pre-trained knowledge is preserved. The network that preserves more pre-trained knowledge is considered more reliable and thus temporarily assumes the guiding role (*i.e.,* teacher), while the other acts as the learner (*i.e.,* student). This dynamic role assignment ensures effective collaboration and adaptability throughout the training process. In addition, we introduce the *direct knowledge distillation* to align the predictions of the student network with those of the teacher, enabling the transfer of domain-specific knowledge at each training iteration. We also employ the *reverse knowledge distillation* loss to encourage diversity by slightly deviating the teacher from the student, preventing both networks from collapsing into identical representations. In this way, we achieve a balance between exploration (via reverse distillation) and exploitation (via direct alignment) and ensure the networks maintain complementary strengths. To further improve feature robustness, we propose a triplet relationship loss that models the relationships between the teacher, student, and the pre-trained SAM.

In summary, our contributions are as follows:

- We propose a Collaborative Mutual Learning framework for the source-free domain adaptation of SAM. Our framework dynamically assigns roles to collaborative networks based on their reliability and encourages mutual learning during training.

- We introduce the direct alignment, reverse distillation, and triplet relationship losses to ensure robust and efficient adaptation.

- Extensive experiments demonstrate the effectiveness of our framework in adapting SAM to challenging target domains, including medical imaging, camouflaged object detection, and robotic vision.

## 2. Related work

### 2.1. Image Segmentation

Image segmentation is a fundamental problem in the field of visual perception (Cao et al., 2023; Zhou et al., 2023b; Jin et al., 2023; Wu et al., 2024; Zhao & Tao, 2023). Over the years, research in image segmentation has evolved from small, task-specific models to large foundational models. Mask R-CNN (He et al., 2017) is a widely used two-stage instance segmentation model that first detects the regions of foreground instances and then generates their segmentation masks. In contrast, YOLACT (Bolya et al., 2019) is a single-stage approach that directly outputs segmentation masks from image feature maps. With advancements in transformers and pre-training techniques, large visual models have garnered significant attention from researchers. The Segment Anything Model (SAM) (Kirillov et al., 2023) is the first visual foundational model to achieve remarkable segmentation performance. SAM is capable of handling various types of prompts, such as bounding boxes, points, polygons, and text descriptions, and produces fine-grained segmentation masks. By pre-training on massive and diverse datasets, SAM exhibits outstanding zero-shot generalization ability, making it applicable to a wide range of tasks. Similarly, DINO v2 (Oquab et al., 2023) leverages self-supervised learning to acquire knowledge from large datasets and demonstrates significant performance improvements on many downstream tasks. These breakthroughs highlight the growing potential of pre-trained foundational models in advancing segmentation tasks.

### 2.2. Domain Adaptation

Domain adaptation (DA) is a critical technique for addressing domain shifts between the source (training) and target (testing) domains. Many methods have been proposed to tackle this problem effectively (Liu et al., 2023b; Chen et al., 2018; Liu et al., 2024b; Chen et al., 2022; Liu et al., 2024a). DANN (Ganin & Lempitsky, 2015) employs generative adversarial learning with a Gradient Reversal Layer to achieve significant improvements in unsupervised domain adaptation. Recent methods like SIGMA (Li et al., 2022) and CIGAR (Liu et al., 2023a) transform image features into

graph space and align category-level graph node features across domains. TENT (Wang et al., 2020) minimizes the entropy loss of model predictions during testing to handle domain shifts. Similarly, MLFA (Liu et al., 2024a) addresses object detection problems by modeling and aligning the feature distributions of the source and target domains. In the context of source-free domain adaptation, SHOT (Liang et al., 2021) utilizes self-training with pseudo-labels generated for target domain data, significantly improving performance without requiring source data. DePT (Gao et al., 2022) adopts prompt learning to address source-free domain adaptation by fine-tuning only the visual prompts from the source domain. WDASS (Das et al., 2023) proposes a weakly supervised method to tackle unsupervised domain adaptation problems. Wang et al. propose the Continual Test-Time Domain Adaptation (CTTA) framework (Wang et al., 2022), which incrementally adapts a source model to continually evolving target domains. Note (Gong et al., 2022) introduces instance-aware batch normalization to address the normalization challenges posed by new target domain samples. RMT (Döbler et al., 2023) employs symmetric cross-entropy loss within a mean teacher framework to tackle the CTTA problem effectively. For SAM adaptation, WeSAM (Zhang et al., 2024) introduces a weakly supervised self-training method. However, it employs a fixed teacher-student architecture, which limits its ability to adapt effectively in challenging target domains.

## 3. Motivation

Existing single-model adaptation methods often struggle to balance the preservation of pre-trained knowledge with the adaptation to target-specific knowledge. Recent approaches using knowledge distillation, such as teacher-student networks (Zhang et al., 2024), have shown a potential to address this challenge. This method has a significant limitation: the roles of teacher and student networks are fixed throughout the training process, based on the assumption that the teacher network consistently provides more reliable predictions than the student network. However, this assumption is often unrealistic when adapting to entirely different target domains. In such cases, both networks may fail to provide accurate predictions at different stages of training, limiting the effectiveness of the adaptation process.

**Our Approach: Collaborative Mutual Learning.** To address these limitations, we propose a novel framework based on mutual learning, where two independent networks collaborate and dynamically learn from each other. This approach overcomes the restrictions of fixed teacher-student methods and offers several key advantages: 1) **Collaborative Knowledge Sharing**. Mutual learning enables two networks to exchange information during training, leading to a more comprehensive understanding of the target domain.

Each network brings a unique perspective, allowing them to complement one another and collaboratively explore the characteristics of the target domains. 2) **Dynamic Role Assignment**. Unlike fixed teacher-student architectures, our method dynamically determines which network acts as the *teacher* and *student* at each iteration. This decision is based on the reliability of their real-time predictions. The more reliable network guides the less reliable one, ensuring that the learning process remains adaptive and responsive to the changing reliability of predictions. By leveraging these advantages, our mutual learning framework achieves a more effective and robust source-free adaptation of SAM.

## 4. Preliminary: Segment Anything Model

The Segment Anything Model (SAM) (Kirillov et al., 2023) is a state-of-the-art foundational model designed for visual perception tasks. Its architecture follows an encoder-decoder framework and comprises three key components: an image encoder $E_{img}$, a prompt encoder $E_{prompt}$, and a mask decoder $D_{mask}$. Given an input image $I$ and associated prompts Prompt (such as bounding boxes, points, masks, or textual descriptions), the image encoder generates an image feature map $F$, and the prompt encoder produces a prompt embedding $P$: $F = E_{img}(I), P = E_{prompt}(\text{Prompt})$. The mask decoder then processes both $F$ and $P$ to produce a segmentation score map $S$: $S = D_{mask}(F, P)$. Finally, a binarization threshold $\mathbb{T}$ is applied to the score map $S$ to produce the segmentation mask $M$. Although SAM demonstrates strong performance across various segmentation tasks, its direct application to target domains with significant distribution shifts often results in suboptimal performance (Zhang et al., 2024; Chen et al., 2023; 2024).

## 5. Method

In this section, we present our framework for source-free domain adaptation of SAM, which utilizes a Collaborative Mutual Learning strategy. We first describe the overall architecture in Sec. 5.1. Then, we detail the Mutual Learning and Dynamic Role Assignment mechanism in Sec. 5.2. Finally, we introduce the optimization objectives in Sec. 5.3.

### 5.1. Architecture

The proposed framework leverages a dual-network architecture, as illustrated in Fig. 1. It consists of a frozen SAM and two independent collaborative image encoders, denoted as $E_{img}^{\alpha}$ and $E_{img}^{\beta}$. Each collaborative encoder is built by extending the SAM image encoder with a LoRA-based fine-tuning module for efficient adaptation (Hu et al., 2022). To encourage diversity and avoid symmetrical failures, the collaborative encoders are initialized with slight parameter

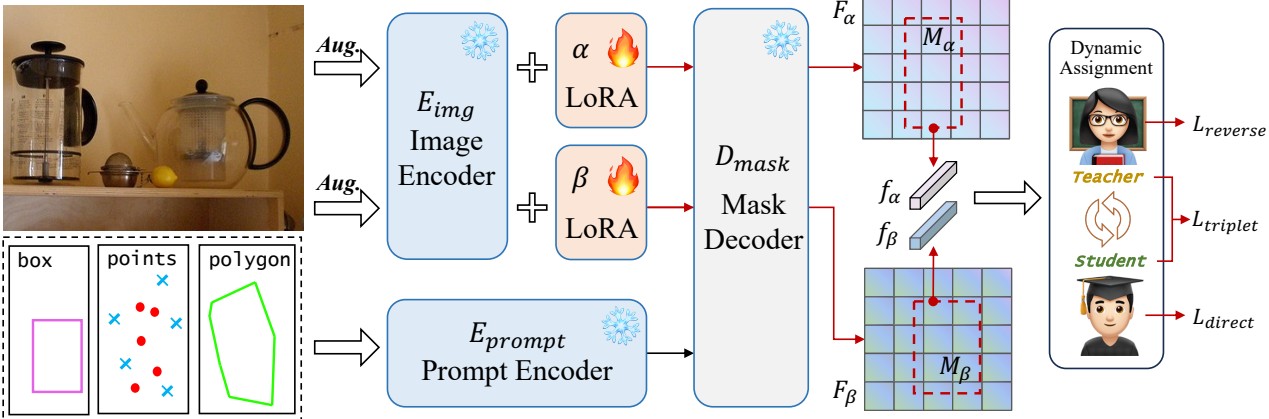

*Figure 1.* Overview of the proposed Collaborative Mutual Learning framework for source-free domain adaptation of SAM. It consists of the pre-trained SAM and two collaborative image encoders ($E_{img}^{\alpha}$ and $E_{img}^{\beta}$). Each collaborative image encoder is built from the original SAM image encoder and a LoRA module. Our proposed mutual learning strategy dynamically assigns the *teacher* and *student* roles to the two cooperative networks during each training iteration. We optimize these cooperative networks using the direct distillation loss, the reverse distillation loss, and the triplet relationship loss.

perturbations from the pre-trained SAM. Specifically, the collaborative encoders are defined as:

$$E_{img}^{\alpha/\beta} = \mathbf{perturb}(E_{img}) + \triangle E_{img}^{\alpha/\beta}, \quad (1)$$

where $\Delta E_{img}^{\alpha/\beta}$ represents the low-rank weights introduced by LoRA and $\mathbf{perturb}(\cdot)$ denotes the parameter perturbation operation. This design enables efficient fine-tuning with minimal additional parameters while retaining the knowledge of the pre-trained SAM image encoder.

During training, each collaborative encoder processes a differently augmented view of the input image $I$, denoted as $I_{\alpha}^{Aug}$ and $I_{\beta}^{Aug}$. The image augmentations (*e.g.,* , random color, brightness, contrast, shadows) expose the collaborative encoders to diverse visual patterns, encouraging them to learn complementary representations of the target domain. The resulting feature maps are: $F_{\alpha/\beta} = E_{img}^{\alpha/\beta}(I_{\alpha/\beta}^{Aug})$, where $F_{\alpha/\beta} \in \mathbb{R}^{h \times w \times c}$, and $h$, $w$, and $c$ denote the height, width, and channel dimensions of the feature map.

We adopt the same method as WeSAM (Zhang et al., 2024) to generate prompts (Prompt). Specifically, five positive points within the ground truth mask and five negative points outside it are randomly selected as point prompts. Polygon prompts are generated by fitting coarse polygons around the ground truth masks. The shared mask decoder integrates image features and prompt embeddings to produce a segmentation score map: $S_{\alpha/\beta} = \text{Sigmoid}[D_{mask}(F_{\alpha/\beta}, P)]$. The score map $S_{\alpha/\beta}$ is binarized with a threshold $\mathbb{T}$ to produce the final segmentation mask: $M_{\alpha/\beta} \in {0, 1}^{h \times w}$. These components form two independent collaborative SAM networks:

$$\mathbf{SAM}_{\alpha/\beta} = D_{mask}[E_{img}^{\alpha/\beta}(I_{\alpha/\beta}^{Aug}), E_{prompt}(\mathsf{Prompt})]. \quad (2)$$

By fine-tuning only the collaborative image encoders via LoRA, the framework achieves efficient and domain-specific adaptation while preserving the rich representations of the pre-trained SAM. At the end of training, the better-performing network is selected as the adapted target SAM model.

## 5.2. Mutual Learning and Dynamic Role Assignment

The core innovation of our framework is the mutual learning mechanism, which dynamically assigns the roles of teacher ($\mathcal{T}$) and student ($\mathcal{S}$) to the two collaborative SAM networks ($\mathbf{SAM}_{\alpha}$ and $\mathbf{SAM}_{\beta}$). Different from the fixed-role knowledge distillation, this dynamic role assignment allows the networks to guide each other alternatively.

### A. Reliability Estimation

In the adaptation process, the pre-trained models often forget their knowledge due to overfitting. This may degrade the reliability of the models. Normally, more knowledge preserved means higher reliability (Zhang et al., 2024). To this end, we extract the foreground representations from the image feature maps and corresponding pseudo-segmentation masks of two collaborative networks ($\mathbf{SAM}_{\alpha/\beta}$) and a reference network ($\mathbf{SAM}_{\gamma}$, original pre-trained SAM). Given the segmentation score map $S_{\alpha/\beta/\gamma}$ and image feature map $F_{\alpha/\beta/\gamma}$ generated by these three networks, the foreground representation $f_{\alpha/\beta/\gamma}$ is computed as the mean feature vectors of pixels within the predicted foreground region:

$$f_{\alpha/\beta/\gamma} = \frac{\sum_{i,j}^{h,w} F_{\alpha/\beta/\gamma}^{i,j} \cdot \mathbf{1}(S_{\alpha/\beta/\gamma}^{i,j} > \mathcal{T})}{\sum_{i,j}^{h,w} \mathbf{1}(S_{\alpha/\beta/\gamma}^{i,j} > \mathcal{T})}, \quad (3)$$

where $F_{\alpha/\beta/\gamma}^{i,j} \in \mathbb{R}^c$ represents the feature vector at pixel

$(i, j)$, and $\mathbf{1}(\cdot)$ is an indicator function that identifies foreground pixels based on the threshold $\mathbb{T}$. Given a foreground, we assume that the cooperative network that preserves more knowledge produces a more similar feature map to the reference network. So we calculate the cosine similarity between the foreground representation of each collaborative network ($f_\alpha$ or $f_\beta$) and that of the pre-trained reference SAM ($f_\gamma$):

$$sim_\alpha = \frac{f_\alpha \cdot f_{ref}}{\|f_\alpha\| \|f_{ref}\|}, \quad sim_\beta = \frac{f_\beta \cdot f_{ref}}{\|f_\beta\| \|f_{ref}\|}. \quad (4)$$

Although $f_\alpha$ remains frozen throughout, it encodes rich and reliable pre-trained knowledge learned from large-scale datasets. The similarity to $f_\alpha$ measures how much pre-trained knowledge is preserved during adaptation. Collaborative Networks with higher similarity to $f_\alpha$ are considered more reliable, as their updates are more cautious and less prone to overfitting to noisy or domain-specific features. So we consider the network with the higher similarity score to be the teacher, while the other becomes the student:

$$(\mathcal{T}, \mathcal{S}) = \begin{cases} (\alpha, \beta) & \text{if } sim_\alpha > sim_\beta, \\ (\beta, \alpha) & \text{otherwise.} \end{cases} \quad (5)$$

This dynamic assignment mechanism ensures that the roles are updated adaptively in each iteration based on the reliability of networks, enabling them to learn from each other effectively.

**B. Role Assignment Distillation**

Once the teacher and student roles are assigned, knowledge distillation is performed to share information between the collaborative networks while preserving their complementary strengths. This is achieved through the direct alignment, the reverse distillation, and the triplet relationship losses.

**Direct Alignment via Knowledge Distillation**. The student network is trained to align its predictions with those of the teacher network, enabling the transfer of task-specific knowledge. The direct alignment loss minimizes the difference between their segmentation score maps:

$$L_{direct} = \begin{cases} \mathcal{L}_{SAM}(x : S_\alpha, y : S_\beta) & (\mathcal{T}, \mathcal{S}) = (\beta, \alpha) \\ \mathcal{L}_{SAM}(x : S_\beta, y : S_\alpha) & (\mathcal{T}, \mathcal{S}) = (\alpha, \beta) \end{cases} \quad (6)$$

Similarly to (Zhang et al., 2024) and (Carion et al., 2020), the SAM loss $\mathcal{L}_{SAM}$ consists the focal loss Focal (Ross & Dollár, 2017) and the Dice loss Dice (Milletari et al., 2016):

$$\mathcal{L}_{SAM}(x, y) = \text{Focal}(x, \text{detach}(y)) + \text{Dice}(x, \text{detach}(y)). \quad (7)$$

The detach operation $\text{detach}(\cdot)$ ensures that the target score map (from the teacher) does not participate in backpropagation, providing stable supervision for the student network.

**Reverse Distillation for Network Diversity**. Inspired by (Zhou et al., 2023a; 2022), we introduce a reverse distillation loss to avoid over-convergence and ensure diversity

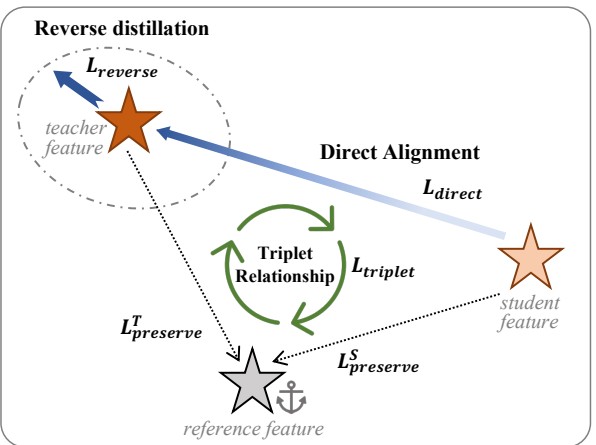

*Figure 2.* The instruction of the loss components in our proposed collaborative mutual learning method.

between two collaborative networks. Different from direct alignment, this loss encourages the teacher network to deviate slightly from the predictions of the student, preventing both networks from collapsing into identical representations. The loss also expands the solution space of the teacher network, avoiding local minima caused by over-alignment with the student. The reverse distillation loss is defined as:

$$L_{reverse} = \begin{cases} -\mathcal{L}_{SAM}(x : S_\beta, y : S_\alpha) & (\mathcal{T}, \mathcal{S}) = (\beta, \alpha), \\ -\mathcal{L}_{SAM}(x : S_\alpha, y : S_\beta) & (\mathcal{T}, \mathcal{S}) = (\alpha, \beta). \end{cases} \quad (8)$$

Similar to (Zhang et al., 2024), to prevent catastrophic forgetting of the pre-trained SAM, we also constrain the outputs of both collaborative networks to remain close to the reference predictions of SAM. This is achieved by minimizing the SAM loss between each collaborative network and the reference:

$$L_{preserve}^{\mathcal{T}/\mathcal{S}} = \begin{cases} \mathcal{L}_{SAM}(x : S_{\beta/\alpha}, y : S_\gamma) & (\mathcal{T}, \mathcal{S}) = (\beta, \alpha), \\ \mathcal{L}_{SAM}(x : S_{\alpha/\beta}, y : S_\gamma) & (\mathcal{T}, \mathcal{S}) = (\alpha, \beta), \end{cases} \quad (9)$$

**Triplet Relationship Loss**. To further enhance the robustness of feature learning, we employ a triplet relationship loss to model the relationships between the teacher, student, and the reference foreground features. During each training iteration, the foreground representations of the teacher ($f_\mathcal{T}$), student ($f_\mathcal{S}$), and reference ($f_\gamma$) are extracted following Eq. (3). The triplet relationship loss is defined as:

$$L_{triplet} = \max(0, \|f_\mathcal{S} - f_\gamma\|_2^2 - \|f_\mathcal{T} - f_\gamma\|_2^2 + \delta), \quad (10)$$

where $\delta > 0$ is a margin hyperparameter. This loss ensures that the teacher features remain closer to the reference features while pushing the student features further away. This loss can maintain the feature consistency across the pre-trained SAM, teacher, and student networks. Preventing excessive divergence and overfitting ensures meaningful

feature relationships and works synergistically with reverse distillation and direct alignment losses to enable robust optimization.

## 5.3. Model Optimization

The total optimization loss is defined as:

$$
\begin{aligned}
L_{total} = L_{direct} &+ \lambda_{reverse}L_{reverse} + \lambda_{triplet}L_{triplet} \\
&+ L_{preserve}^{\mathcal{S}} + (1 + \lambda_{preserve})L_{preserve}^{\mathcal{T}},
\end{aligned}
\tag{11}
$$

where $\lambda_{reverse}$, $\lambda_{presercve}$, and $\lambda_{triplet}$ are weights of reverse distillation, knowledge preserve, and triplet relationship losses, respectively. $\lambda_{reverse}$ is set to a smaller value, while $\lambda_{preserve}$ exerts a larger weight on $L_{preserve}$ to improve the stability of teacher network training. As shown in Fig. 2, the gradients from $L_{distill}$, $L_{reverse}$, and $L_{triplet}$ encourage convergence toward complementary yet distinct networks, while $L_{preserve}$ preserve pre-trained knowledge of original SAM.

# 6. Experiment

## 6.1. Datasets and Evaluation

To evaluate the effectiveness of our proposed method, we conduct extensive experiments on four types of downstream segmentation tasks, including natural images, medical images, camouflaged images, and robotic images. These tasks exhibit varying degrees of domain shift from the pre-trained dataset of SAM. We used the Mean Intersection over Union (mIoU) as the primary metric to assess adaptation performance on specific target domains. The results of all comparative methods were obtained from (Zhang et al., 2024). To provide a more comprehensive evaluation of each adaptation method, we calculated the *average mIoU*, which reflects the mean performance of the method across three types of prompts (box, points, and polygon) used for image segmentation. Additionally, we introduced the *GAIN* metric to quantify the performance improvement of each adaptation method over the original SAM (source model). There are six benchmark datasets employed in our experiments: COCO (Lin et al., 2014), Pascal VOC (Everingham et al., 2015), Kvasir-SEG (Jha et al., 2020), CAMO (Le et al., 2019), COD10K (Fan et al., 2020), and OCID (Suchi et al., 2019).

**COCO.** COCO is a widely used dataset in computer vision tasks, including object detection, segmentation, keypoint detection, and image captioning. It contains over 330,000 images and 200,000 labeled images. It provides pixel-level instance segmentation labels of 80 categories.

**Pascal VOC.** Pascal VOC is a dataset composed of natural images, featuring 20 object categories with corresponding bounding box and pixel-level labels. We utilize its 2012 version as the target domain, which includes 11,530 images and 27,450 labeled instances.

**Kvasir-SEG.** Kvasir-SEG is a medical image segmentation dataset containing a large number of gastrointestinal endoscopy images. It provides pixel-level segmentation masks for diseased areas, such as gastrointestinal polyps. This dataset is used as the target domain for medical-specific segmentation tasks.

**CAMO.** CAMO is a dataset for camouflaged object detection. It primarily contains images of camouflaged animals and artificially hidden objects. The dataset includes 1,000 images with pixel-level segmentation masks. We employ CAMO to evaluate the adaptation performance when transferring from natural images to camouflaged scenarios.

**COD10K.** COD10K is a widely used dataset for camouflaged object detection. It contains 10,000 images categorized into five super-categories and 69 sub-categories. The camouflaged objects include animals, plants, artificial objects, and more. This dataset provides a diverse evaluation of methods for detecting camouflaged objects.

**OCID.** OCID is a dataset designed for object detection and segmentation in indoor scenes, with a focus on recognizing objects in high-density stacking scenarios. The images are collected from various indoor environments to simulate object stacking and occlusion challenges encountered during robotic operations. OCID contains over 20,000 images across 89 categories.

## 6.2. Implementation Details

The collaborative encoder $E_{img}^{\alpha/\beta}$ is initialized using the pre-trained SAM image encoder with a ViT-B backbone. To prevent symmetrical failures in the collaborative image encoders, we set the amplitude of the normally distributed random noise **perturb** to 0.001. During the adaptation process, we freeze the SAM model and utilize LoRA to fine-tune only a small subset of parameters, enabling efficient adaptation. We use the Adaptive Moment Estimation (Adam) optimizer (Kingma, 2014) with a learning rate of 0.0001 and a weight decay of 0.0001. The model is trained for 20,000 iterations on each target domain, with a batch size of 1. The hyperparameters $\lambda_{reverse}$ and $\lambda_{preserve}$ in the total optimization loss are set to 0.1. To balance the training between the teacher and student networks, $\lambda_{triplet}$ is set to a small value of 0.01. All experiments are conducted using four NVIDIA A100 GPUs.

## 6.3. Comparison with State-Of-The-Arts

**Adaptation to natural target domains.** Tab. 1 presents the comparison results on downstream target domains with natural images. Our method achieves an average mIoU of 71.50 on COCO and 76.45 on Pascal VOC, outperforming existing methods. Compared with WeSAM, WDASS, and SHOT, our approach improves the average mIoU on COCO

*Table 1.* **Comparison Results on COCO and Pascal VOC.** *Source* and *Target* denote the models trained with source domain data and target domain data, respectively. WeSAM* denotes reproduced results of WeSAM.

| Method | COCO | | | | | Pascal VOC | | | | |
|---|---|---|---|---|---|---|---|---|---|---|
| | box | point | poly | Average | GAIN | box | point | poly | Average | GAIN |
| Source | 74.29 | 55.06 | 65.64 | 65.00 | - | 69.21 | 69.21 | 60.79 | 66.40 | - |
| Target | 81.50 | 69.77 | 73.39 | 74.89 | 9.90 | 81.23 | 76.98 | 71.32 | 76.51 | 10.11 |
| TENT (Wang et al., 2020) | 78.21 | 52.99 | 71.51 | 67.57 | 2.58 | 80.24 | 74.97 | 65.03 | 73.41 | 7.01 |
| SHOT (Liang et al., 2021) | 75.18 | 58.46 | 69.26 | 67.63 | 2.64 | 79.80 | 74.26 | 63.38 | 72.48 | 6.08 |
| soft Teacher (Xu et al., 2021) | 75.94 | 43.36 | 68.27 | 62.52 | -2.47 | 72.93 | 56.09 | 62.20 | 63.74 | -2.66 |
| TRIBE (Su et al., 2024) | 77.56 | 49.56 | 70.99 | 66.04 | 1.05 | 78.87 | 69.21 | 65.39 | 71.16 | 4.76 |
| DePT (Gao et al., 2022) | 71.00 | 37.35 | 63.27 | 57.21 | -7.78 | 74.09 | 42.99 | 59.94 | 59.01 | -7.39 |
| WDASS (Das et al., 2023) | 77.29 | 60.55 | 70.19 | 69.34 | 4.35 | 80.12 | 76.15 | 66.98 | 74.42 | 8.02 |
| WeSAM* (Zhang et al., 2024) | 77.32 | 60.50 | 70.77 | 69.53 | 4.54 | 80.27 | 74.15 | 66.72 | 73.71 | 7.31 |
| ours | **78.97** | **63.00** | **72.54** | **71.50** | **6.51** | **82.90** | **76.24** | **70.20** | **76.45** | **10.05** |

*Table 2.* **Comparison Results on CAMO and COD10K.** *Source* and *Target* denote the models trained with source domain data and target domain data, respectively.

| Method | CAMO | | | | | COD10K | | | | |
|---|---|---|---|---|---|---|---|---|---|---|
| | box | point | poly | Average | GAIN | box | point | poly | Average | GAIN |
| Source | 62.72 | 57.43 | 50.85 | 57.00 | - | 66.32 | 63.61 | 40.04 | 56.66 | - |
| Target | 79.17 | 77.01 | 67.12 | 74.43 | 17.43 | 78.06 | 78.44 | 64.90 | 73.80 | 17.15 |
| TENT (Wang et al., 2020) | 71.24 | 59.59 | 60.29 | 63.71 | 6.71 | 69.36 | 61.94 | 43.36 | 58.22 | 1.57 |
| SHOT (Liang et al., 2021) | 71.61 | 62.78 | 58.72 | 64.37 | 7.37 | 69.09 | 65.25 | 42.38 | 58.91 | 2.26 |
| soft Teacher (Xu et al., 2021) | 62.30 | 48.64 | 51.26 | 54.07 | -2.93 | 66.32 | 50.04 | 32.27 | 49.54 | -7.11 |
| TRIBE (Su et al., 2024) | 66.00 | 61.97 | 60.54 | 62.84 | 5.84 | 67.84 | 63.62 | 42.75 | 58.07 | 1.42 |
| DePT (Gao et al., 2022) | 55.44 | 33.07 | 48.63 | 45.71 | -11.29 | 59.32 | 34.06 | 35.51 | 42.96 | -13.69 |
| WDASS (Das et al., 2023) | 71.25 | 63.39 | 62.29 | 65.64 | 8.64 | 71.42 | 65.61 | 43.93 | 60.32 | 3.67 |
| WeSAM (Zhang et al., 2024) | 73.42 | 65.55 | 62.90 | 67.29 | 10.29 | 71.93 | 70.55 | 45.87 | 62.78 | 6.13 |
| ours | **74.46** | **70.21** | **67.54** | **70.74** | **13.74** | **73.89** | **72.83** | **47.27** | **64.66** | **8.01** |

by 1.97, 2.16, and 3.87, respectively. On Pascal VOC, our method surpasses WeSAM by 2.74 and WDASS by 2.03 in average mIoU. These results highlight the effectiveness of our approach in adapting to challenging scenarios.

**Adaptation to camouflaged target domains.** As illustrated in Tab. 2, we report the performance of various state-of-the-art methods on the COCO and COD10K datasets. Our method achieves the highest average mIoU of 70.74 on COCO and 64.66 on COD10K. Specifically, compared to the self-training method WeSAM, our approach delivers substantial improvements of 3.45 and 1.88 in average mIoU on CAMO and COD10K, respectively. In terms of GAIN, our method also achieved the largest improvement 13.74 on CAMO and 8.01 on COD10K. Additionally, when compared with other methods like WDASS and SHOT, our method maintains consistent superiority across all metrics, further demonstrating its robustness and versatility.

**Adaptation to medical target domains.** We summarize the comparison results on kvasir-SEG in Tab. 3. Our method achieves an average mIoU of 83.29, significantly outperforming all competing methods. Compared with WeSAM, TRIBE, and WDASS, our approach improves the average mIoU by 7.26, 5.06, and 12.43, respectively. Our method achieves a remarkable GAIN improvement of 17.32, sur-

passing WeSAM by 7.26 and TRIBE by 9.06, demonstrating its clear superiority.

**Adaptation to robotic target domains.** Tab. 4 lists the results on the OCID dataset. The segmentation results using polygon prompts are significantly better than those obtained with box or point prompts. These results might be caused by the sparse supervision provided by box and point prompts, which often includes multiple overlapping or similar instances. Such cases result in ambiguous boundaries and imprecise masks. In contrast, polygon prompts explicitly define instance boundaries, reducing ambiguity and enabling more accurate segmentation. Our method achieves an average mIoU of 84.02, outperforming all competing methods. Compared with WeSAM and WDASS, our approach surpasses them by 2.14 and 3.52 in average mIoU. Additionally, our method achieves a GAIN of 7.16, which is significantly higher than that of SHOT (2.71) and TRIBE (0.19). These results demonstrate the effectiveness and robustness of our method in adapting to the target domain.

### 6.4. Qualitative Results

Fig. 3 presents the qualitative results of our proposed method alongside the comparison methods, SAM and We-SAM. The visualizations demonstrate that our method

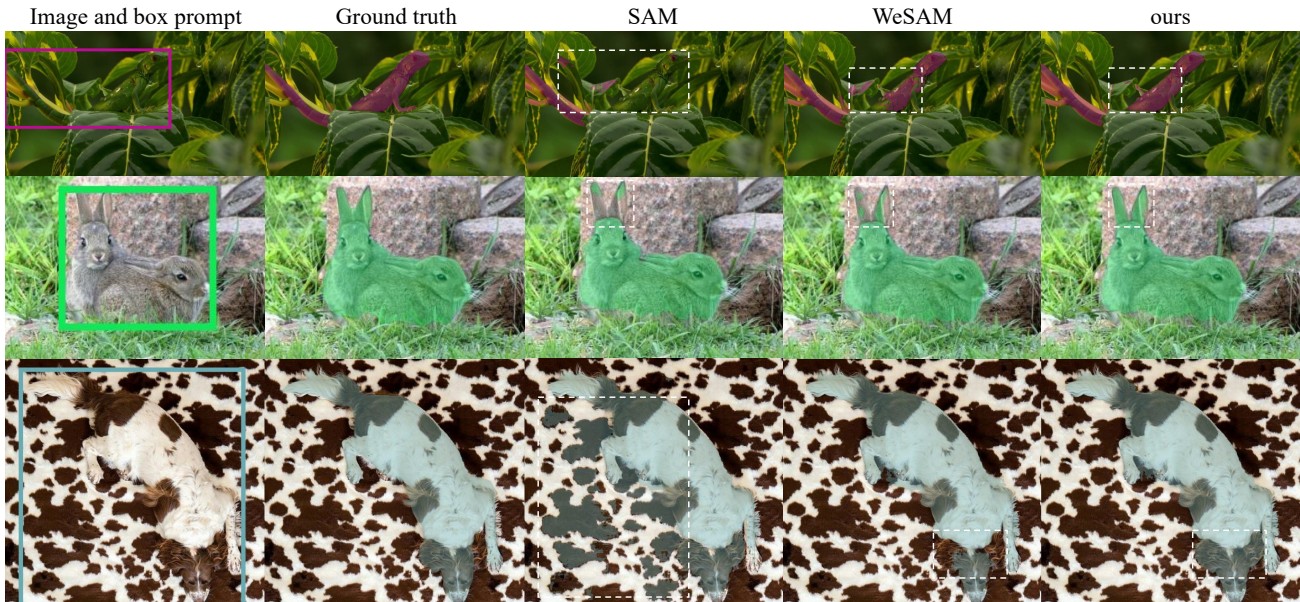

*Figure 3.* Qualitative comparison results on the CAMO target domain among SAM, WeSAM, and our proposed method.

*Table 3.* **Comparison Results on kvasir-SEG.** *Source* and *Target* denote the models trained with source domain data and target domain data, respectively.

| Method | kvasir-SEG | | | | |
| --- | --- | --- | --- | --- | --- |
| | box | point | poly | Average | GAIN |
| Source | 81.59 | 62.30 | 54.03 | 65.97 | - |
| Target | 85.89 | 77.54 | 81.64 | 81.69 | 15.72 |
| TENT | 82.47 | 61.84 | 62.97 | 69.09 | 3.12 |
| SHOT | 82.30 | 63.76 | 61.34 | 69.13 | 3.16 |
| soft Teacher | 84.12 | 73.53 | 58.15 | 71.93 | 5.96 |
| TRIBE | 85.05 | 73.03 | 64.61 | 74.23 | 8.26 |
| DePT | 81.91 | 52.06 | 61.55 | 65.17 | -0.80 |
| WDASS | 84.01 | 63.78 | 64.78 | 70.86 | 4.89 |
| WeSAM | 85.47 | 75.23 | 67.40 | 76.03 | 10.06 |
| ours | **86.92** | **76.18** | **86.78** | **83.29** | **17.32** |

*Table 4.* **Comparison Results on OCID.** *Source* and *Target* denote the models trained with source domain data and target domain data, respectively.

| Method | OCID | | | | |
| --- | --- | --- | --- | --- | --- |
| | box | point | poly | Average | GAIN |
| Source | 86.35 | 71.41 | 72.81 | 76.86 | - |
| Target | 91.24 | 89.22 | 79.23 | 86.56 | 9.71 |
| TENT | 87.77 | 66.61 | 77.53 | 77.30 | 0.45 |
| SHOT | 88.06 | 74.39 | 76.25 | 79.57 | 2.71 |
| soft Teacher | 84.98 | 68.46 | 73.75 | 75.73 | -1.13 |
| TRIBE | 86.77 | 67.86 | 76.50 | 77.04 | 0.19 |
| DePT | 82.00 | 56.52 | 70.92 | 69.81 | -7.04 |
| WDASS | 87.68 | 77.13 | 76.70 | 80.50 | 3.65 |
| WeSAM | **88.09** | **80.14** | 77.41 | 81.88 | 5.02 |
| ours | 88.07 | 77.33 | **86.66** | **84.02** | **7.16** |

*Table 5.* **Performance comparison of our proposed method with different loss components.** *Direct*, *Reverse*, and *Triplet* denote the direct alignment distillation loss, reverse distillation loss, and triplet relationship loss, respectively.

| Direct | Reverse | Triplet | CAMO | COD10K |
| --- | --- | --- | --- | --- |
| ✓ | | | 68.83 | 63.21 |
| ✓ | ✓ | | 70.22 | 64.13 |
| ✓ | | ✓ | 69.75 | 63.75 |
| ✓ | ✓ | ✓ | 70.74 | 64.66 |

achieves finer and more precise segmentation performance in the downstream segmentation task.

### 6.5. Ablation Studies

To evaluate the effectiveness of each component in our proposed method, we conduct ablation studies on the adaptation tasks from source to CAMO and COD10K. We employ the Average mIoU as the metric for evaluating their adaptation performance.

**Ablation on the reverse distillation loss.** Tab. 5 shows the impact of the Reverse Distillation loss on adaptation performance. By comparing the baseline method with only Direct loss (68.83 on CAMO and 63.21 on COD10K) to the method with both Direct and Reverse losses (70.22 on CAMO and 64.13 on COD10K), it is evident that the Reverse loss significantly enhances performance. The improvements of 1.39 on CAMO and 0.92 on COD10K demonstrate that the Reverse Distillation loss boosts network diversity, enabling the

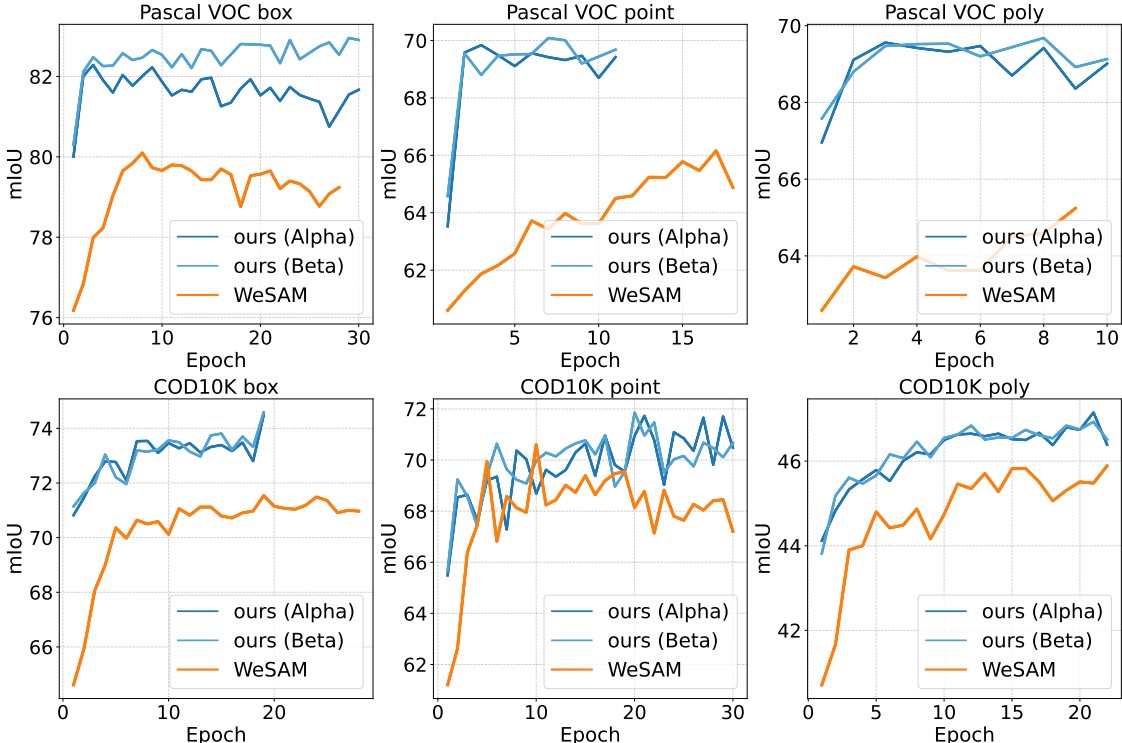

*Figure 4.* **Segmentation performance curves of our method and WeSAM.** The experiments were conducted in different prompt forms on two target domain datasets. The orange, blue, and light blue curves represent the performances of WeSAM, $SAM_\alpha$, and $SAM_\alpha$ in our method, respectively.

model to learn more robust and complementary features for better adaptation results.

**Ablation on the triplet relationship loss.** Tab. 5 highlights the contribution of the Triplet Relationship loss to adaptation performance. Adding the Triplet Relationship loss to the baseline with Direct loss achieves performance gains from 68.83 to 69.75 on CAMO and from 63.21 to 63.75 on COD10K. These improvements (0.92 on CAMO and 0.54 on COD10K) show that the Triplet loss effectively captures the triplet relationships, leading to enhanced feature representation and improved adaptation performance.

### 6.6. Effectiveness and Efficiency Analysis

Fig. 4 illustrates the mIoU performance curves of our proposed dual collaborative-network method (Ours) and the baseline method (WeSAM) on two datasets. Our method exhibits significantly faster performance improvement compared to WeSAM on both datasets. The mIoU curves of $SAM_\alpha$ and $SAM_\beta$ in our method show alternating improvements, demonstrating the mutual learning process. This allows our model to achieve higher performance levels in fewer epochs. Furthermore, the final performance of both $SAM_\alpha$ and $SAM_\beta$ surpasses that of WeSAM, highlighting the effectiveness of collaborative mutual learning. The

dynamic teacher-student role switching and collaborative mutual learning in our method lead to faster convergence and superior performance, validating its efficiency and effectiveness over fixed-role approaches.

## 7. Conclusion

We propose a Collaborative Mutual Learning framework for source-free domain adaptation of the Segment Anything Model (SAM). Our approach introduces a dual-network structure and facilitates their interaction through a flexible and adaptive training process. During the adaptation process, we estimate the reliability of the two collaborative networks and propose a Dynamic Role Assignment mechanism to dynamically assign *teacher* and *student* roles to the networks. To enhance learning, we incorporate the direct alignment loss, enabling the student network to effectively learn domain-specific features from the teacher. Additionally, we employ the reverse distillation loss to avoid over-convergence and ensure the diversity of the teacher network. Furthermore, we design a triplet relationship loss to encourage robust feature learning while preserving the pre-trained knowledge of SAM. Extensive experiments conducted across various challenging target domains demonstrate the effectiveness and adaptability of our framework.

## Acknowledgment

This research is supported by Laboratory for Artificial Intelligence in Design (Project Code: RP3-3) under InnoHK Research Clusters, Hong Kong SAR Government.

## Impact Statement

This paper presents work whose goal is to enhance the performance of foundational models within the target domain. There are many potential societal consequences of our work, none which we feel must be specifically highlighted here.

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
