# OpenReview forum: "Mutual Learning for SAM Adaptation: A Dual Collaborative Network Framework for Source-Free Domain Transfer"
_ICML.cc/2025/Conference — ICML 2025 poster_

### Official Review · Reviewer_B25v · 2025-03-07

**Overall Recommendation:** 5

**Summary:**

The paper proposes a Collaborative Mutual Learning Framework for the source-free domain adaptation of the Segment Anything Model. The key innovation lies in dynamically assigning "teacher" and "student" roles to two collaborative networks based on their reliability during training. The framework introduces three key components direct alignment loss for knowledge transfer, reverse distillation loss for diversity, and triplet relationship loss. Extensive experiments demonstrate superior performance across several challenging target domains, including medical imaging, camouflaged object detection, and robotic vision.

**Claims And Evidence:**

1. The proposed framework achieves state-of-the-art results on multiple benchmarks.
2. The dynamic teacher-student assignment is validated through performance curves and ablation studies.

**Essential References Not Discussed:**

No major references appear to be missing, but the authors could consider citing additional works on dynamic teacher-student frameworks or mutual learning approaches in other fields to further contextualize the novelty of their method.

**Experimental Designs Or Analyses:**

The experiments are comprehensive, covering a wide range of domains and providing ablation studies to validate the contributions of individual components.

**Methods And Evaluation Criteria:**

The methods are well-designed, and the evaluation criteria are appropriate for the task. The use of standard datasets like COCO and Pascal VOC ensures comparability with prior work. The inclusion of metrics like mIoU and GAIN is appropriate for assessing segmentation performance.

**Other Comments Or Suggestions:**

Improve the clarity of certain equations, particularly the triplet loss formulation, which could be made more intuitive.

**Other Strengths And Weaknesses:**

This work presents a range of strengths.
1. The dynamic teacher-student assignment is innovative and addresses the limitations of fixed teacher-student frameworks.
2. The paper evaluates the framework on six datasets across diverse domains, demonstrating its versatility and robustness.
3. The use of LoRA for efficient fine-tuning is a practical and computationally efficient choice.
4. The framework achieves state-of-the-art performance, consistently outperforming existing methods like WeSAM and SHOT on mIoU and GAIN metrics.
5. The method is highly relevant for real-world applications, where source data is often inaccessible due to privacy restrictions.

However, I am still concerned about the following questions.
1. The triplet loss contributes only marginal improvements, raising questions about its necessity.
2. The method is heavily tailored to SAM, and its generalizability to other foundational models is unclear.
3. The paper does not discuss the sensitivity of the framework to hyperparameters (e.g., thresholds for role assignment, loss weights), which could impact reproducibility.

**Questions For Authors:**

1. Could the dynamic role-switching mechanism be extended to more than two networks, or is it inherently limited to a dual-network setup?
2. The supplementary material shows the performance of different role selection schemes. Should the original SAM data be added to the table, so as to show the various methods more intuitively?

**Relation To Broader Scientific Literature:**

The paper is well-situated in the domain adaptation literature and builds on prior work like WeSAM and SHOT. However, the connection to broader mutual learning frameworks outside SAM adaptation could be strengthened.

**Theoretical Claims:**

There are no complex theoretical claims in this work. The reliability metric and its use in dynamic role assignment are intuitive.

---

> ### Author Rebuttal · Authors · 2025-04-01
>
> Thank you for your critical as well as constructive assessments of our work, and we address the concerns point by point as follows.
>
> ---
>
> ### **W1: The triplet loss contributes only marginal improvements, raising questions about its necessity.**
>
> The triplet loss, although contributing marginally to numerical performance, plays a critical theoretical role in stabilizing the training process. It ensures feature consistency between the pre-trained SAM, teacher, and student networks, preventing excessive divergence that could lead to noisy predictions or unstable optimization. From a theoretical perspective:
>
> 1. The triplet loss acts as a regularizer, preserving the reference model’s general-purpose knowledge during adaptation.
>
> 2. It prevents overfitting by maintaining meaningful feature relationships across networks.
>
> 3. Its synergy with reverse distillation and direct alignment losses ensures robust training.
>
> While its numerical gains may appear small, its role in stabilizing the mutual learning process is essential. This theoretical significance will be clarified in the revised manuscript.
>
> ### **W2: The method is heavily tailored to SAM, and its generalizability to other foundational models is unclear.**
>
> Our framework is not inherently tied to SAM and can be generalized to other foundational models. Theoretical principles such as dynamic teacher-student assignment, reverse distillation, and triplet loss are broadly applicable:
>
> 1. **Dynamic Role Assignment:** Reliance on real-time reliability scoring allows adaptation to any pre-trained model.
>
> 2. **Reverse Distillation:** Promotes diversity and avoids overfitting, critical for models fine-tuned on new domains.
>
> 3. **Triplet Loss:** Ensures consistent alignment with the foundational model while adapting to target domains.
>
> We will explicitly discuss this generalizability and propose extending the framework to other models as future work.
>
> ### **W3: The framework’s sensitivity to hyperparameters (e.g., thresholds for role assignment, loss weights) is not discussed, which could impact reproducibility.**
>
> We conducted additional experiments analyzing the impact of key hyperparameters.
>
> 1. **Threshold for Role Assignment:** The values of $\mathcal{T}$ around 0.5 consistently yielded the best or second-best results across datasets, confirming robustness to moderate variations.
>
> |$\mathcal{T}$|0.3|0.4|0.5|0.6|0.7|
> |-|-|-|-|-|-|
> |source$\to$CAMO|66.59|70.36|70.74|70.13|69.92|
> |source$\to$COD10K|60.12|63.74|64.66|64.85|64.27|
>
> 2. **Loss Weights:** Reverse distillation and direct alignment losses were set to the same weight based on their design purpose. Optimal weights were determined as 0.1 for reverse and direct alignment losses and 0.001 for the triplet loss. Extremely high or low weights resulted in significant performance degradation. These values were validated across datasets and generalized to all experiments.
>
> |$\lambda_{reverse}$|$\lambda_{preserve}$|$\lambda_{triplet}$|CAMO|COD10K|
> |-|-|-|-|-|
> |0.02|0.02|0.001|69.23|63.45|
> |0.1|0.1|0.01|70.74|64.66|
> |0.1|0.1|0.05|70.32|64.37|
> |0.5|0.5|0.1|68.81|62.93|
>
> We will include these findings in the revised manuscript and supplementary material to enhance reproducibility.
>
> ### **W4: The triplet loss formulation could be made more intuitive.**
>
> To improve clarity, we will revise the explanation of the triplet loss in the manuscript with step-by-step derivations and an intuitive diagram illustrating its role in aligning feature representations between the reference, teacher, and student networks. This will help readers better understand its purpose and implementation.
>
> ### **Q1: Could the dynamic role-switching mechanism be extended to more than two networks?**
>
> Our current framework is designed for two networks to simplify the reliability-based role assignment process, as it allows for a straightforward comparison of reliability scores. However, the mechanism can be extended to more than two networks by generalizing the reliability metric to rank multiple networks and dynamically assign roles (e.g., multiple teachers and students). This extension could potentially enhance performance further by leveraging a broader ensemble of models. While we have not implemented this in the current work, we consider it an exciting direction for future research and will discuss this possibility in the revised manuscript.
>
> ### **Q2: Should the original SAM data be added to the table in the supplementary material to make the role selection schemes more intuitive?**
>
> We agree with this suggestion. Adding the original SAM data will provide a clearer baseline for understanding the improvements achieved by our role selection schemes. We will update the supplementary material to include this data and highlight the relative gains of each scheme.
>
> |role assignment strategy|CAMO|COD10K|
> |-|-|-|
> |source(SAM)|57.00|56.66|
> |WeSAM|67.29|62.78|
> |random|60.97|50.22|
> |reverse|53.61|38.32|
> |ours|70.74|64.66|

---

> > ### Comment · Reviewer_B25v · 2025-04-05
> >
> > The authors have provided a rebuttal that addresses my main concerns.  Their explanation of the analysis of the triplet loss, the experiments of hyperparameters, and the framework’s generalizability adds clarity to the paper.  After reading the paper and the authors' rebuttal, I think their response is reasonable for those questions, particularly regarding computational performance trade-offs.  Experiments demonstrate that the proposed method achieves significant performance gains with minimal additional computational cost, as the dual LoRA adapter only slightly increases training time and memory while outperforming the baseline.  So I support the acceptance of this paper and will improve my recommendation to Strong Accept.

---

> > > ### Author Response · Authors · 2025-04-06
> > >
> > > We sincerely thank you for your thoughtful evaluation and valuable feedback on our work. Your encouraging comments and constructive suggestions mean a great deal to us and have provided significant motivation to further improve the quality of our research. We deeply appreciate the time and effort you have invested in reviewing our manuscript and offering insightful recommendations. We will carefully revise the manuscript, addressing all reviewers’ comments to ensure clarity. Thank you again for your valuable time and support!

---

### Official Review · Reviewer_s4SY · 2025-03-11

**Overall Recommendation:** 4

**Summary:**

This paper addresses the challenge of adapting the Segment Anything Model to new domains with significant distribution shifts through a Collaborative Mutual Learning Framework. The method involves two collaborative networks that dynamically alternate between teacher and student roles based on reliability. They also propose the reverse distillation to train the teacher network in each training iteration. Experiments on six datasets demonstrate the approach's adaptability and superior performance compared to existing methods.

**Claims And Evidence:**

The claims are backed by strong experimental evidence. The authors demonstrate state-of-the-art results on multiple datasets and provide ablation studies to validate each proposed component.

**Essential References Not Discussed:**

The authors have cited relevant works in SAM adaptation, domain adaptation, and mutual learning.

**Experimental Designs Or Analyses:**

The experimental designs are robust. The ablation studies justify the contributions of each loss term, and the comparisons with state-of-the-art methods are thorough.

**Methods And Evaluation Criteria:**

The methods are reasonable and align well with the problem's requirements. The use of standard benchmarks like COCO and Pascal VOC provides credibility.

**Other Comments Or Suggestions:**

(1) The framework does not explicitly analyze how often roles are switched or how this impacts convergence and performance.
(2) The dynamic teacher-student process is not visualized, making it difficult to understand how roles evolve during training.
(3) The paper lacks detailed visualizations of segmentation improvements, which would strengthen the qualitative evaluation.

**Other Strengths And Weaknesses:**

Strengths:
(1) Dynamic teacher-student assignment is an innovative idea.
(2) The paper provides detailed ablation experiments to validate the contribution of each loss component.
(3) The introduction of the GAIN metric provides an intuitive measure of performance improvement over the baseline SAM.
(4) The combination of direct alignment, reverse distillation, and triplet losses is well-designed to balance exploration and exploitation.

**Questions For Authors:**

Could you provide more theoretical insights or empirical evidence to justify why reverse distillation enhances diversity?

**Relation To Broader Scientific Literature:**

The paper is well-situated within the literature on SAM and source-free domain adaptation. It builds on previous works like WeSAM but innovates by introducing a dynamic mutual learning framework.

**Theoretical Claims:**

The paper does not seem to make any complex theoretical claims requiring proof.

---

> ### Author Rebuttal · Authors · 2025-04-01
>
> Thank you for your critical as well as constructive assessments of our work, and we address the concerns point by point as follows.
>
> ---
>
> ### **W1: Lack of analysis on role switching frequency and its Impact on convergence and performance**
>
> We conducted an additional quantitative analysis of the role-switching frequency across training iterations. Specifically, we tracked the number of role switches for both networks during training on three datasets. The results show that the switching frequency decreases as training progresses, stabilizing after several epochs. This behavior aligns with our design goal: early in training, frequent switches allow both networks to explore the target domain, while later, more stable role assignments aid convergence.
>
> |Target|total iteration|1|2|3|4|5|
> |-|-|-|-|-|-|-|
> |CAMO|1972|976|638|416|204|103|
> |OCID|1000|469|491|466|457|197|
> |Kvasir-SEG|836|444|375|305|29|10|
>
> We also evaluated the impact of role switching on performance by comparing source SAM, a fixed-role method (WeSAM), and our dynamic switching framework. The dynamic role assignment consistently outperformed the fixed-role method. These results confirm that dynamic switching is crucial for effective domain adaptation.
>
> |role assignment strategy|CAMO|COD10K|
> |-|-|-|
> |source(SAM)|57.00|56.66|
> |WeSAM(fixed)|67.29|62.78|
> |random(mutual)|60.97|50.22|
> |reverse(mutual)|53.61|38.32|
> |ours(mutual)|70.74|64.66|
>
> ### **W2: Lack of visualization of the dynamic Teacher-Student Process**
>
> We generated a visualization of the reliability scores for both networks over training iterations. This visualization demonstrates how the reliability-based role assignment mechanism dynamically adjusts roles based on the real-time performance of each network. Early in training, roles alternate frequently as both networks adapt to the target domain. Over time, the network with consistently higher reliability assumes the teacher role more often, reflecting a stabilization in the mutual learning process. This visualization will be included in the revised manuscript or open-source platform to illustrate how roles evolve during training and to provide a clearer understanding of the dynamic teacher-student process.
>
> ### **W3: Lack of detailed visualizations of segmentation improvements**
>
> We have prepared detailed visualizations of segmentation results on challenging samples from the CAMO, COD10K, and OCID datasets. These visualizations highlight the improvements achieved by our method, particularly in cases involving occlusion, complex object boundaries, and cluttered scenes. For example, on the OCID dataset, our method produces cleaner and more precise segmentation masks compared to both the baseline SAM and WeSAM, especially when using polygon prompts. These improvements are attributed to the combination of our proposed loss functions, which enhance feature alignment, promote diversity, and ensure robust training. We will include these visualizations in the revised manuscript and provide additional examples on the open-source platform for further clarity.
>
>
> ### **Q1: Could you provide more theoretical insights or empirical evidence to justify why reverse distillation enhances diversity?**
>
> The reverse distillation loss is designed to encourage output diversity by promoting a divergence between the teacher and student networks. We summarize its theoretical and empirical justifications:
>
> **Theoretical Insights**:
>
> 1. **Balancing Exploration and Exploitation:** Reverse distillation introduces controlled divergence between the teacher and student, forcing the teacher to explore diverse predictions. This ensures the teacher provides richer and more comprehensive guidance.
>
> 2. **Regularization Effect:** It prevents knowledge collapse by maintaining teacher-student diversity, critical for adapting to domains with significant distribution shifts.
>
> 3. **Optimization Perspective:** This loss expands the teacher’s solution space, avoiding local minima caused by excessive alignment with the student.
>
> 4. **Model Ensemble Parallel:** Similar to ensemble learning, diversity between teacher and student predictions strengthens generalization.
>
> **Empirical Evidence**:
>
> We conducted an ablation study comparing the performance of our method with and without the reverse distillation loss. On the CAMO dataset, removing the reverse distillation loss led to a 0.99 drop in average mIoU, while on COD10K, the performance dropped by 0.91 average mIoU.
>
> |Direct|Reverse|Triplet|CAMO|COD10K|
> |-|-|-|-|-|
> |✓|||68.83|63.21|
> |✓|✓||70.22|64.13|
> |✓||✓|69.75|63.75|
> |✓|✓|✓|70.74|64.66|
>
> In summary, reverse distillation enhances diversity, improves robustness, and synergizes with other loss terms to ensure effective knowledge transfer. These insights will be added to the revised manuscript.

---

> > ### Comment · Reviewer_s4SY · 2025-04-03
> >
> > The authors’ responses address my concerns. The analysis of role-switching frequency and its impact on performance demonstrates the benefits of dynamic role assignment. The visualizations are important for a comprehensive evaluation of the method’s performance, especially in challenging scenarios, and I encourage their inclusion in the final manuscript. Additionally, the justification for reverse distillation is supported by ablation studies. Therefore, I am revising my recommendation to accept.

---

> > > ### Author Response · Authors · 2025-04-03
> > >
> > > We sincerely appreciate the reviewer’s careful reconsideration and willingness to upgrade the score to accept based on our rebuttal. Your insights and constructive feedback significantly helped us improve the clarity and quality of our manuscript. We will ensure all suggestions are thoroughly incorporated into the final version of the paper.

---

### Official Review · Reviewer_t7Bb · 2025-03-12

**Overall Recommendation:** 3

**Summary:**

This paper introduces a mutual teaching framework for SAM adaptation in target domains. Compared to self-training framework, the mutual learning enables dynamic assignment of teacher and student roles, leading to more robust and generalized performance. To facilitate adaptation, three loss functions are proposed: direct alignment loss, reverse distillation loss, and triplet relationship loss. Extensive experiments demonstrate the effectiveness of the proposed framework.

**Claims And Evidence:**

Line 256 states, “Similar to (Zhang et al., 2024), to prevent catastrophic preserving of the pre-trained SAM.” Please review this claim carefully, especially the misleading term "catastrophic preservation". WeSAM proposes regularization on the student model using an anchor model to prevent model collapse. The proposed preseving loss is to prevent catastrophic forgetting of the pretrained knowledge, which is conceptually different from preventing catastrophic preservation of SAM.

**Essential References Not Discussed:**

No

**Experimental Designs Or Analyses:**

1. The experiments do not clearly demonstrate how WeSAM would perform if adapted to a mutual learning framework, making it difficult to attribute the improvements solely to the proposed modifications.
2. In Table 4, the proposed model performs worse than the baseline WeSAM with box and point prompts, but shows significant improvement with the polygon prompt. The authors should provide an analysis of the reasons behind this difference.

**Methods And Evaluation Criteria:**

This paper largely adopts the model architecture, loss design motivations, and experimental settings from the baseline WeSAM.

**Other Comments Or Suggestions:**

No other comments.

**Other Strengths And Weaknesses:**

Strengths:
1. The paper is well-writen and clear.
2. The experiments are sufficient to evaluate the effectiveness of the proposed model.

Weaknesses:
1. Model design. From my understanding, the proposed method improves upon the baseline WeSAM by replacing the self-training framework with a mutual teaching framework. However, it is well-known that mutual learning is inherently more robust to noise and better generalizes to diverse target distributions, which naturally leads to improved performance. While this paper inherits these advantages of mutual learning, it also retains some of its limitations, such as high computational and design complexity.
2. Loss design. The direct alignment loss, reverse distillation loss, and triplet loss are designed based on motivations presented in WeSAM.
Therefore, the improved performance over WeSAM can mainly be attributed to mutual learning. Beyond this, I find limited insights into the model architecture and loss design motivations.

**Questions For Authors:**

Please correct me if I have any misunderstandings.

**Relation To Broader Scientific Literature:**

1. This paper primarily adopts the model architecture and loss design motivations from the existing work WeSAM, including: 1) the model architecture with a fixed encoder network, prompt encoder, and mask decoder; 2) fine-tuning the LoRA module for efficient SAM adaptation; 3) a teacher-student self-training loss; 4) a regularization loss from the fixed anchor/reference model to prevent model collapse; and 5) a regularization loss in the feature space. Additionally, the specific loss formulations (i.e., the SAM loss) are consistent with existing work, while the triplet loss is derived from metric learning and domain adaptation papers.

2. The mutual teaching framework is based on concepts from many existing semi-supervised learning and domain adaptation papers.

3. The triplet loss is derived from existing metric learning and domain adaptation papers.

**Theoretical Claims:**

There are no specific theoretical claims or corresponding proofs, but the overall model design is reasonable.

---

> ### Author Rebuttal · Authors · 2025-04-01
>
> Thank you for your critical as well as constructive assessments of our work, and we address the concerns point by point as follows.
>
> ---
>
> ### **Claims And Evidence: Clarification on "Catastrophic Preservation" in line 256**
>
> We appreciate your observation regarding the term "catastrophic preservation". In the revised manuscript, we will rephrase this term to "preventing catastrophic forgetting" to align with standard terminology and clarify its conceptual basis.
>
> ### **Experimental 1: Attribution of improvements to the mutual learning framework**
>
> We conduct an ablation study to evaluate the impact of different role assignment strategies (learning paradigms) in our framework. Experimental results are as follows. These results confirm that the performance improvements are directly attributable to our dynamic role assignment mechanism in mutual learning.
>
> |role assignment strategy|CAMO|COD10K|
> |-|-|-|
> |source(SAM)|57.00|56.66|
> |WeSAM(fixed)|67.29|62.78|
> |random(mutual)|60.97|50.22|
> |reverse(mutual)|53.61|38.32|
> |ours(mutual)|70.74|64.66|
>
> 1. **Role Assignment Matters:** Random or reverse assignments disrupt learning, leading to performance drops, while our dynamic role assignment achieves the best results by leveraging real-time reliability evaluation.
>
> 2. **Reverse Assignment Impact:** Incorrectly assigning the less reliable model as the teacher severely hampers knowledge transfer, highlighting the importance of proper role allocation.
>
> 3. **Dynamic Role Assignment Effectiveness:** Our approach ensures stable collaboration and effective knowledge transfer, significantly outperforming fixed or random assignments.
>
> We also counted the number of roles exchanged at different training epochs in several adaptation tasks:
>
> |Target|total iteration|1|2|3|4|5|
> |-|-|-|-|-|-|-|
> |CAMO|1972|976|638|416|204|103|
> |OCID|1000|469|491|466|457|197|
> |Kvasir-SEG|836|444|375|305|29|10|
>
> The results align with our design goal: early in training, frequent switches allow both networks to explore the target domain, while later, more stable role assignments aid convergence.
>
> ### **Experimental 2: Performance differences with Prompts (Table 4)**
>
> OCID images have dense objects with severe occlusion and overlap, making segmentation difficult. Sparse supervision from box and point prompts often includes multiple overlapping or similar instances, leading to ambiguous boundaries and imprecise masks. Polygon prompts explicitly describe instance boundaries, reducing ambiguity and enabling more accurate segmentation.
>
> WeSAM’s fixed teacher provides stable guidance, which is beneficial when sparse prompts introduce noise or ambiguity. Our dynamic role assignment can amplify noise in challenging scenarios, leading to suboptimal role allocations. WeSAM may be better tuned for sparse prompts, while our framework emphasizes bidirectional learning.
>
> ### **W1: Mutual learning is inherently more robust but comes with higher computational and design complexity, which is a limitation of the proposed method.**
>
> We conducted experiments comparing the training time and memory usage and performance of WeSAM, a single LoRA adapter, and our dual LoRA adapter framework on the source$\to$Kvasir-SEG(box) task. The results are as follows:
>
> ||WeSAM|single adapter|dual adapter (ours)|
> |-|-|-|-|
> |training time per iteration (s)|**1.8**|1.1|1.3|
> |CUDA memory (MB)|12500|14276|**14906**|
> |mIoU|85.47|-|**86.92**|
>
> Our method achieves a balance between computational efficiency and performance. Compared to the single LoRA adapter, the dual adapter only increases training time by 0.2 seconds per iteration and requires 4.4% more memory, while delivering significantly better results. Importantly, our algorithmic optimizations make our method faster than WeSAM (1.3s vs. 1.8s per iteration) despite incorporating a more advanced mutual learning framework.
>
> ### **W2: Insights into loss design motivations**
>
> Our loss functions are designed to enhance the effectiveness of the mutual teaching framework by facilitating knowledge transfer, increasing model diversity, and stabilizing training.
>
> 1. **Direct Alignment Loss:** This loss aligns the student network’s predictions with the teacher’s outputs, enabling effective transfer of target domain knowledge from the teacher to the student. It ensures the student learns domain-adapted features from the teacher.
>
> 2. **Reverse Distillation Loss:** Unlike direct alignment, this loss increases the divergence between the teacher and student networks, encouraging the teacher to explore more diverse outputs. By providing the teacher with greater optimization freedom, this loss enhances the robustness of the teacher network.
>
> 3. **Triplet Relationship Loss:** This loss reinforces the feature relationships among the reference model (pre-trained SAM), the teacher, and the student. By aligning their relative feature representations, ensures consistent and robust training, improving target domain adaptation.

---

> > ### Comment · Reviewer_t7Bb · 2025-04-05
> >
> > Thank you for the rebuttals to my reviews. However, several key concerns remain unaddressed—particularly regarding the model and loss design. 1) For W1: While I appreciate the inclusion of computational cost results, I was hoping for a clearer explanation of how your method differs from the baselines, especially WeSAM. As mentioned in my review, your model seems to follow WeSAM closely in terms of architecture, loss design, and experimental setup. Since mutual learning is already known to be more robust to noise and generalize better, it seems the main change is simply replacing the teacher-student setup with mutual learning. 2) For W2: I was not asking for the meaning of each loss. What I wanted to understand is how your loss design brings something new or different compared to WeSAM. 3) For E2: The authors acknowledged that WeSAM’s fixed teacher offers more stable guidance, especially with sparse prompts, while your model may amplify noise in such cases. This suggests the challenges of applying mutual teaching in this setting remain unresolved. The proposed method seems to leverage known strengths of mutual learning without addressing its limitations.
> >
> > If I’ve misunderstood any of these points, I would appreciate clarification.

---

> > > ### Author Response · Authors · 2025-04-07
> > >
> > > We deeply appreciate your detailed comments. Below, we address your concerns point by point.
> > >
> > > ### **W1: How does our method fundamentally differ from WeSAM?**
> > >
> > > Our method replaces the fixed teacher-student paradigm in WeSAM with a dynamic mutual learning framework. This change introduces the following fundamental differences:
> > >
> > > 1. Unlike WeSAM’s fixed teacher-student setup, our framework dynamically assigns teacher and student roles based on real-time reliability estimation. This mechanism ensures that the more reliable model provides guidance, while the less reliable model continues to learn, effectively adapting to dynamic shifts in training. Our dynamic role assignment mechanism builds on ideas from [1] but extends them to source-free adaptation with SAM. Unlike prior work, we introduce a reliability-based metric tailored for segmentation tasks.
> > >
> > > 2. While WeSAM uses a single teacher-student pair, we employ two collaborative networks that simultaneously learn from each other. Each network explores complementary features of the target domain, enhancing robustness and generalization.
> > > 3. We introduce a triplet relationship loss to preserve the pre-trained SAM knowledge while enhancing robustness during adaptation. In contrast, WeSAM relies solely on anchor-based regularization. Our triplet loss aligns the relative feature representations of the teacher, student, and pre-trained model, stabilizing training and improving feature consistency across domains.
> > > 4. Despite the dual LoRA adapter framework, our method achieves better computational efficiency compared to WeSAM (**1.3s vs. 1.8s** per iteration with NVIDIA A100) due to algorithmic optimizations. Memory usage increases by only **4.4%**, while delivering significant performance gains (e.g., **Kvasir-SEG: 86.92 mIoU vs. 85.47 for WeSAM**). It is mainly achieved through the following aspects:
> > > - Lightweight Training: Each LoRA adapter adds only **36,864×r parameters (r=4: 147K)** to the frozen SAM encoder. Dual adapters **total <0.3M parameters, consuming <5%** extra VRAM compared to single-model training.
> > > - Optimized Computation: Backpropagation is only applied to LoRA weights (**not SAM's 600M+ parameters**)
> > >
> > > ### **W2: Novelty in loss design compared to WeSAM**
> > >
> > > WeSAM primarily utilizes a teacher-student loss for pseudo-labeling and a regularization loss to preserve pre-trained knowledge. In contrast, our framework introduces novel loss functions tailored specifically for mutual learning:
> > >
> > > 1. Direct Alignment Loss
> > >
> > > While WeSAM relies on static teacher-student alignment, this loss dynamically assigns teacher and student roles based on real-time reliability. This bidirectional alignment ensures optimal knowledge transfer by always leveraging the stronger network, a feature absent in WeSAM.
> > >
> > > 2. Reverse Distillation Loss
> > >
> > > This loss is uniquely designed to foster diversity by encouraging the teacher and student networks to deviate slightly from each other. This prevents over-convergence and promotes complementary learning, enhancing robustness to domain shifts. Ablation studies confirm that removing reverse loss leads to significant performance degradation, demonstrating its vital role.
> > >
> > > 3. Triplet Relationship Loss
> > >
> > > Unlike WeSAM’s anchor-based regularization, this loss explicitly models feature relationships among the pre-trained SAM, teacher, and student. This ensures consistent knowledge preservation while enhancing target domain adaptation. Inspired by metric learning, this loss is specifically adapted to the mutual learning context, where maintaining pre-trained knowledge is critical.
> > >
> > > ### **E2: Challenges with Sparse Prompts and Noise Amplification**
> > >
> > > Our dynamic role assignment relies on real-time reliability estimation, which could inadvertently propagate noise if the reliability metric is affected by ambiguous or noisy pseudo-labels. This explains why WeSAM performs better with sparse prompts in Table 4. To address this limitation, we are exploring strategies to make the dynamic role assignment more robust to sparse prompts. These include integrating additional confidence metrics into the reliability estimation process or incorporating a noise-aware regularization term. We believe such enhancements could mitigate the noise amplification issue while preserving the adaptability of mutual learning.
> > >
> > > Despite these challenges in OCID, it is important to note that our method achieves significantly better performance across all other datasets. **The results in Tables 1, 2, and 3 demonstrate its effectiveness across diverse target domains.** These results highlight the broader applicability and potential of our method in handling substantial domain shifts.
> > >
> > > **Please feel free to reach out with any further questions or feedback. We deeply value your insights and remain committed to addressing any remaining concerns. Thank you for your time and consideration.**
> > >
> > > [1] Zhou L, et al. Adaptive mutual learning for unsupervised domain adaptation[J]. IEEE TCSVT, 2023.

---

### Official Review · Reviewer_SWqC · 2025-03-13

**Overall Recommendation:** 3

**Summary:**

The paper introduces a Collaborative Mutual Learning Framework for source-free domain adaptation of the Segment Anything Model. It employs dual networks that alternate roles as teacher and student based on reliability, preserving SAM’s pre-trained knowledge. T Experiments in medical, camouflaged, and robotic domains show state-of-the-art performance, outperforming WeSAM and SHOT in mIoU.

**Claims And Evidence:**

The paper's motivation (L121-L125) states that previous works using the teacher-student approach assume that the teacher network consistently provides more reliable predictions than the student network. However, I believe this is not the actual assumption. In my opinion, the teacher model is updated by the student model using EMA, which helps better prevent catastrophic forgetting, while the student model's purpose is to explore a broader space to guide the teacher model in the right direction.

**Essential References Not Discussed:**

NO

**Experimental Designs Or Analyses:**

The proposed method achieves state-of-the-art performance on various benchmarks. However, I would like to know how to set $\mathcal{T}$ in Eq. (3). I hope the authors can include an ablation study on this parameter. Additionally, does $\mathcal{T}$ need to be set to different values for different domains?

**Methods And Evaluation Criteria:**

Make sense.

**Other Comments Or Suggestions:**

None

**Other Strengths And Weaknesses:**

See below (Questions For Authors*).

**Questions For Authors:**

1. How is $f_{\gamma}$ obtained? Is it derived from the raw pre-trained SAM encoder without any LoRA modifications?

2. Why does computing similarity with $f_{\gamma}$ allow the estimation of model confidence and consequently determine whether A or B should be the teacher? This is particularly unclear if the model corresponding to $f_{\gamma}$ is frozen.

**Relation To Broader Scientific Literature:**

Related to the CTTA setting, which is also a variant of SFDA, many methods in CTTA[1-4] have utilized the Teacher-Student Network. Moreover, the CTTA task also needs to address issues such as error accumulation and catastrophic forgetting, which are also the focus of this paper.

[1] Continual Test-Time Domain Adaptation

[2] Robust mean teacher for continual and gradual test-time adaptation

[3] Exploring sparse visual prompt for cross-domain semantic segmentation

[4] Note: Robust continual test-time adaptation against temporal correlation

**Theoretical Claims:**

I think this paper do not contain the theoretical claims.

---

> ### Author Rebuttal · Authors · 2025-04-01
>
> Thank you for your critical as well as constructive assessments of our work, and we address the concerns point by point as follows.
>
> ---
>
> ### **Claims And Evidence: About the motivation**
>
> We appreciate your detailed understanding of teacher-student networks. Our motivation is not directed at the use of EMA or the exploration role of the student network but rather at the fixed-role assignment in traditional teacher-student frameworks. These methods assume that the teacher consistently provides more reliable predictions than the student, and this fixed-role assumption limits their adaptability.
>
> To address this, we propose a dynamic role assignment mechanism, where teacher and student roles are reassigned based on real-time reliability, enabling more flexible and adaptive knowledge transfer. Additionally, we emphasize robust knowledge distillation by comparing each network's output with the frozen pre-trained SAM. The network retaining more pre-trained knowledge is assigned as the teacher, ensuring cautious and stable updates. This approach reduces error accumulation and catastrophic forgetting.
>
> We will revise the motivation section to clarify these points and include references to related CTTA methods to better contextualize our contributions.
>
> ### **Experimental Designs Or Analyses: About the threshold $\mathcal{T}$ in Eq. (3)**
>
> To set the base value for this threshold, we referred to common practices in SAM applications, where thresholds of 0.5 or 0.7 are often used. To further refine this selection, we conducted cross-validation experiments on domains with significant distribution shifts, specifically the CAMO and COD10K datasets. We present more ablation study experiments on $\mathcal{T}$ in the following table:
>
> |$\mathcal{T}$|0.3|0.4|0.5|0.6|0.7|
> |-|-|-|-|-|-|
> |source$\to$CAMO|66.59|70.36|70.74|70.13|69.92|
> |source$\to$COD10K|60.12|63.74|64.66|64.85|64.27|
>
> We observed that 0.5 consistently produced the best or near-optimal results across both datasets, balancing precision and recall. To ensure fairness and consistency across domains, we used 0.5 as the default threshold for all experiments. It generalizes well, achieving significant performance improvements across diverse datasets without domain-specific tuning. We will include these results and their analysis in the revised manuscript to justify the threshold selection and its impact on performance.
>
> ### **Q1: How is $f_{\gamma}$ obtained?**
>
> Yes, $f_{\gamma}$ is directly derived from the frozen pre-trained SAM encoder, without any LoRA modifications. It represents the general knowledge learned during SAM’s large-scale pre-training. By using a frozen reference, we ensure that $f_{\gamma}$ provides a stable baseline for evaluating how much pre-trained knowledge the collaborative networks retain.
>
> ### **Q2: Why does computing similarity with $f_{\gamma}$ estimate model confidence, especially if $f_{\gamma}$ is frozen?**
>
> The similarity to $f_{\gamma}$ measures how much pre-trained knowledge is preserved during adaptation. Although $f_{\gamma}$ remains frozen throughout, it encodes rich and reliable pre-trained knowledge learned from large-scale datasets. Using $f_{\gamma}$ as a reference allows us to measure how much pre-trained knowledge each network retains during adaptation. Networks with higher similarity to $f_{\gamma}$ are considered more reliable, as their updates are more cautious and less prone to overfitting to noisy, domain-specific features.
>
> This approach has two advantages:
>
> 1. Compared to relying on the teacher network determined in the previous step, which may have already drifted from the pre-trained knowledge, $f_{\gamma}$ provides a stable and consistent baseline for evaluating reliability.
>
> 2. It is more computationally efficient, as it avoids the need for iterative teacher updates. By dynamically assigning the teacher role to the network closer to $f_{\gamma}$, we ensure robust and cautious knowledge transfer, reducing error accumulation and catastrophic forgetting.
>
> Our experiments validate the effectiveness of this method, demonstrating that it achieves stable and efficient domain adaptation while leveraging the reliability of pre-trained knowledge. We will further elaborate on this in the revised manuscript to clarify its rationale and efficiency.
>
> ### **Relation to CTTA methods**
>
> Thank you for pointing out the connection to CTTA methods. While CTTA methods address challenges like error accumulation and catastrophic forgetting, our approach introduces key innovations:
>
> 1. Dynamic Role Assignment based on real-time reliability, allowing flexible teacher-student role switching;
>
> 2. Robust Knowledge Preservation via the frozen pre-trained SAM, preventing overfitting to noisy features;
>
> 3. Bidirectional Knowledge Exchange, enabling mutual learning rather than unidirectional transfer.
>
> We will expand the related work section to highlight these distinctions and include the provided references([1-4]).

---

### Decision · Program_Chairs · 2025-05-01

**Decision:**

Accept (poster)

**Comment:**

This paper proposes a Collaborative Mutual Learning Framework for source-free domain adaptation of the Segment Anything Model. All four reviewers are positive about this paper, and the rebuttal addresses most of the reviewers' concerns. The AC recommends acceptance.